# Engineering Requirements of a Herpes Simplex Virus Patient Registry: Discovery Phase of a Real-World Evidence Platform to Advance Pharmacogenomics and Personalized Medicine

**DOI:** 10.3390/biomedicines7040100

**Published:** 2019-12-15

**Authors:** Svitlana Surodina, Ching Lam, Caroline de Cock, Michelle van Velthoven, Madison Milne-Ives, Edward Meinert

**Affiliations:** 1Skein LTD, Kemp House, 152-160 City Road, London EC1V 2NX, UK; svitlana@skeingroup.com; 2Department of Paediatrics, University of Oxford, Oxford OX3 9DU, UK; ching.lam@stx.ox.ac.uk (C.L.); caroline.decock@paediatrics.ox.ac.uk (C.d.C.); michelle.vanvelthoven@paediatrics.ox.ac.uk (M.v.V.); madison.milne-ives@paediatrics.ox.ac.uk (M.M.-I.); 3Institute of Biomedical Engineering, Department of Engineering Science, University of Oxford, Oxford OX3 7DQ, UK; 4Department of Primary Care and Public Health, School of Public Health, Imperial College London, London W6 8RP, UK

**Keywords:** registries, herpes simplex, pharmacokinetics, data collection

## Abstract

Comprehensive pharmacogenomic understanding requires both robust genomic and demographic data. Patient registries present an opportunity to collect large amounts of robust, patient-level data. Pharmacogenomic advancement in the treatment of infectious diseases is yet to be fully realised. Herpes simplex virus (HSV) is one disease for which pharmacogenomic understanding is wanting. This paper aims to understand the key factors that impact data collection quality for medical registries and suggest potential design features of an HSV medical registry to overcome current constraints and allow for this data to be used as a complement to genomic and clinical data to further the treatment of HSV. This paper outlines the discovery phase for the development of an HSV registry with the aim of learning about the users and their contexts, the technological constraints and the potential improvements that can be made. The design requirements and user stories for the HSV registry have been identified for further alpha phase development. The current landscape of HSV research and patient registry development were discussed. Through the analysis of the current state of the art and thematic user analysis, potential design features were elucidated to facilitate the collection of high-quality, robust patient-level data which could contribute to advances in pharmacogenomic understanding and personalised medicine in HSV. The user requirements specification for the development of an HSV registry has been summarised and implementation strategies for the alpha phase discussed.

## 1. Introduction

Pharmacogenomics is a mainstay of precision medicine, enabling treatment to be tailored to the genome of the target organism, response to be predicted and likely side effects preempted. The source of pharmacogenomic heterogeneity derives from both pharmacodynamic and pharmacokinetic factors. Pharmacodynamic variability results from differences in the phenotype and genotype of the disease and phenotype of the individual whilst pharmacokinetic variability is influenced by the phenotype of the individual and their lifestyle and environment [1]. To further the use and effectiveness of pharmacogenomics and precision medicine, genomic data must be integrated with population data describing the variations in patient demographics, comorbidities and clinical outcomes. Patient registries are one such method to collate large-scale records of demographic variables and clinical outcomes. These databases could provide data that can be used to identify population sub-groups and account for variables in drug development. New drug treatments could be designed that target specific patient phenotypes [2]. Patient registries could also improve treatment by helping clinicians identify patients with certain risk factors or to target patients with drug regimens that are most likely to be effective and well-tolerated.

Pharmacogenetics is, at present, predominantly employed in oncology; however, has also assumed an important role in infectious disease [3]. *HLAB*5701* has been identified as a biomarker for abacavir hypersensitivity and *c.3435C/T* variation in the *MDR1* gene can be used to predict antiretroviral therapy response [4]. Despite these advances, the role of pharmacogenomics in the improvement of treatment of infectious diseases has yet to be fully realised [5]. Herpes simplex virus (HSV) is a common infectious disease for which pharmacogenomic knowledge is limited. HSV exists as two subtypes, HSV1 and HSV2, with which two-thirds of the global population under 50 years of age are estimated to be infected [6]. HSV is transmitted by intimate personal contact and has the potential to cause significant morbidity, particularly when acquired by neonates during parturition; however, there remains no effective cure or vaccine [7,8]. Patients with HSV are currently treated only when prodromes or symptoms are apparent or are not treated at all. The optimal timing and duration of treatment for HSV are uncertain, and are currently prescribed at doctors’ discretion, necessitating a comprehensive understanding of response to treatment and variability in treatment outcomes to standardise care and optimise outcomes for these patients.

There is an urgent need for a vaccine against HSV2 due to the number of individuals infected globally and the potential resultant morbidity [9], and whilst there are many studies going on in the HSV research domain, researchers who conduct observational studies using existing data are unable to control the data collection and quality [10]. Similarly, the quality of data issues related to the stigma of HSV, privacy concerns, selection bias, and gaps in the understanding of treatments and their outcomes were highlighted as key challenges for researchers in HSV. In the World Health Organisation (WHO)’s roadmap for advancing development of vaccine against sexually transmitted infections (STIs) published in 2016, obtaining better epidemiologic data was highlighted as a key step towards better understanding of STIs, advancing research in vaccine development and expediting the clinical translation of potential vaccines and cures [11,12]. Epidemiologic data would similarly aid in advancing the pharmacogenomic understanding of HSV and the populations that are at high risk of infection. Population-based health care databases may constitute a cost-effective way of conducting epidemiological studies on HSV patients. The large size of the databases offers the potential for precise estimates even when studying rare outcomes or exposures.

The success of drugs designed based on genomic and molecular predictors requires clinical data to be equally precise and reliable [13]. There is, therefore, a clear need for a robust platform for the collation of population-level data. The development of a standardised patient-powered registry tool could further facilitate the collection of high-quality demographic and outcome data that could be compiled with genomic analysis to contribute pharmacogenomic understanding to the evolution of the management of patients with HSV.

This paper describes the discovery phase of a proposed HSV patient registry. The discovery phase is the first phase in agile project delivery, followed by alpha, beta and live phases [14]. The discovery phase involves understanding and defining the problem to be solved, learn about the potential users of the service, constraints that might be encountered and opportunities to improve on the current systems. The alpha and beta phases build upon the findings of the discovery phase; testing out solutions to the problems identified and beginning development in earnest, having identified the most promising solution, respectively. The live phase involves sustaining and maintaining the service with continuous iterations.

This paper describes the background investigation and user analysis that informed the discovery phase of the design of an HSV patient registry, which involves: (1) limitations of current registries, (2) technical standards for the development of medical registries, (3) a medical registries development framework, (4) technical research and planning, (5) user personas and use cases, (6) design requirements for an HSV patient registry, and (7) problem definition. This registry aims to collect patient-level data on the population infected with HSV to aid in the elucidation of pharmacokinetic variables impacting the response to drugs in patients with HSV and ultimately contribute to the development of new precision drug treatments and vaccines for HSV.

## 2. Materials and Methods

The investigation of current medical registries, the standards and frameworks utilised, and their limitations was undertaken to understand the constraints that the development of a novel registry might face and how this registry might improve on current offerings. Existing frameworks and governance surrounding medical registries were also investigated to ensure that the eventual design of the registry adheres to current policy and standards, and aid in technical planning. In-depth structured interviews were conducted with Herpes Virus Association representatives, a clinician and a researcher specialising in HSV to build a thematic analysis of problems related to the availability and quality of HSV data which has been previously reported [15]. The themes identified were used to develop use cases that contributed to the identification of common challenges surrounding registries and implications for system and data collection design. For the present discovery phase, two user personas were chosen: patient and researcher. A third user persona is taken into consideration: System Administrator/Data Scientist. User personas, epics and stories were mapped in detail, specifically understanding the context of patient use cases and researcher data requirements to finalise the details of the motivations and goals of the patient and researcher. Insights from the interviews, use cases and analysis of the current state of the field were used to determine the registry design requirements moving forwards. Finally, the PICO framework was used to analyse the background of the problem and to define the research question for the Alpha phase of the HSV registry.

## 3. Results

### 3.1. Limitations of Current Registries

An exploratory literature search was undertaken to gain insight into current medical registries and their limitations. The United Kingdom is home to over 50 clinical audit programmes, the United States has over 110 federally qualified registries certified to report quality metrics, and Sweden has over 100, covering conditions from birth to old age [16]. A number of limitations in existing registries have been researched and identified. These include privacy concerns, the need for anonymity and informed consent, lack of stakeholder feedback and lack of awareness of existing standards and standard processes when building or maintaining a patient registry [16,17,18,19,20,21]. In addition, there are also examples of limited adoption of registries by certain socio-demographic groups [22] that are linked to the language barriers, level of technology adoption and geographical locations. Low-quality data and a lack of awareness of data standards were also common themes in existing medical registry design approaches [16,20,21].

### 3.2. Technical Standards for the Development of Medical Registries

Registries vary in technical standards used in development. Integration and interoperability of multiple health information systems generally cover multiple layers and aspects of registry operations [20], and technological standards support functional interoperability of such systems. There are a number of standards in operation and development:FHIR is a standard developed by health level seven (HL7) that functions as an application programmer interface (API) for developers to access needed clinical information from the EMR [23]openElectronic Health Record (EHR)EN/ISO 13606—Electronic Health Record CommunicationExtensible Markup Language (XML)The Resource Description Framework (RDF) and RDF-Schema (RDFS)Simple Knowledge Organization System (SKOS)Common Terminology Services, Release 2 (CTS2)

FHIR and openEHR are the two most recent, robust and complete healthcare data persistence and exchange specifications. openEHR and FHIR share many similarities such as data types, the availability of a large number of interoperable and reusable extensions, powerful querying and APIs [24]. However, openEHR, has a complicated health informatics standard with a steep learning curve for the uninitiated. REST is available for openEHR, but with large and complex queries becomes difficult. FHIR leverages a wealth of experience in healthcare interoperability with HL7 Version 2, the predominant interoperability standard used today in medical systems for EMR integration [23] and is easier and cheaper than other comparable standards. It is faster to learn, implement and troubleshoot [25]. A second key advantage of FHIR is the World Wide Web Consortium (W3C) compliant format that follows a language and structure well-established and commonly utilized in the web development community. It is the same format that organizations such as Facebook or Twitter use in their APIs. The structure is called a RESTful architecture which standardizes methods to search for, update, and delete data. Speed, simplicity and widespread use make FHIR an optimal data development format, however, the FHIR standard is still in development [26]. Ultimately, in order to extract data from other platforms, custom interfaces between the registry, and each individual source data system implementation must be created.

### 3.3. Medical Registries Development Framework

We investigated the current development frameworks employed by medical registries to understand their limitations and benefits. Patient-centred registry design was used in a number of registers including The UK Myotonic Dystrophy Patient Registry which is described as “an example of a novel, online-based, cost-effective, and patient-driven registry” [27]. This framework continuously monitors patient outcomes and experiences which is available in real-time to both clinicians and patients to facilitate their joint work [16].

Open-source registry frameworks have been proposed to enable researchers to create their own registries using the RDRF (Rare Disease Registry Framework) [28]. It aims to solve the problem of data integration and facilitate ease of sharing and extensive data mining of patient data across jurisdictional boundaries. The system uses a data element system and derived data elements architecture using the Python programming language to create classes dynamically. The technology stack used includes Python Django, PostgreSQL, JQuery/Bootstrap. Server: Apache + CentOS, MondoDB and combines relational and non-relational data schemas. Bellgard et al. developed the concept on the example of a registry for Gaucher disease which uses Registry Mark-up Language (RML) which illustrates security-focused architecture [29]. These registries utilise SSL and role-based permission access control. The registry framework stores data in PostgreSQL and MongoDB. The databases using these systems are encrypted, including encryption of the communication between the web interface and the database. In terms of physical security, workstations including laptops used to access the registry should require user authorisation, be subject to appropriate security policies, and have appropriate security software installed. On any workstation on which reports may be downloaded from the registry and stored, whole-disk encryption should be implemented on the device to guard against the risk of data exposure through theft or accidental loss. Another example of an open-source based registry based on RDRF—is the global Angelman syndrome registry [30]. Open source registry systems for rare disease use distributed searches to comply with data protection requirements and preserve data sovereignty.

### 3.4. Technical Research and Planning

The final technology stack and tools will be determined after technical, user, and market research is completed. Project requirements such as fast implementation, scalable architecture and use of advanced data science tools and methods for data analysis suggests the use of Python technology on the backend. Some open-source components such as data science modules and user management and admin frameworks will need to be employed. The proposed registry will use API-based REST architecture that will simplify the future extensibility with modules and components such as external EHR services, and third-party connectivity. Security aspects and standards for data collection and storage will be considered to ensure interoperability of the system and integration of the registry with EHR systems. Integration and interoperability are simplified when both the EHR and the registry adhere to modern IT standards for data exchange. The registry data schema design will follow the national standards for electronic health records [15] and the use of pseudo-anonymity techniques will be investigated to maintain patient privacy. This will enable users to access the registry without providing his/her identity. This requires the use of a robust cryptographic hash function to anonymise information related to the patient’s identity and solutions for reversible pseudonym generation. GDPR requirements additionally protect personal data and will be considered during the development of this registry.

Methods for analysis of the data collected by the registry will be defined at later stages of development in line with the objectives of all stakeholders in the data exchange network. These definitions and stakeholder insights regarding current challenges will be used to build a hypothesis around the best ways to implement data science instruments and machine learning to maximise meaningful output from the registry data.

### 3.5. User Personas and Use Cases

In-depth structured interviews were undertaken to elucidate user priorities and challenges with data collection and registries in the field of HSV. The themes and use cases deriving from these interviews have been previously reported [31]. The user challenges and priorities that were identified and the design implications resulting from these themes are outlined in Table 1.

Additionally, patient’s top-level goals and motivations need to be considered, including the desire to learn more about HSV and its subtypes, get cutting edge information on treatments, log recurrence data and help improve scientific knowledge.

For the present discovery phase, two user personas have been chosen: patient and researcher. A third user persona is taken into consideration: System Administrator/Data Scientist. Although not explicitly discussed as a user persona, defining and understanding the needs of the System Administrator is important to make the solution usable, for example, to enable user account management. The researcher user personas and use cases are outlined in Table 2. Based on the use cases, feature requirements were developed, outlined in Table 3.

### 3.6. Design Requirements for an HSV Patient Registry

Based on the research, we have identified the requirements for an HSV medical registry data, technology and user experience design as specified in Table 4.

### 3.7. Problem Definition

The PICO framework was used to analyse the background of the problem and define the research question for the alpha phase of the HSV registry (Table 5) [34].

This discovery phase highlighted the primary research question for the alpha phase of the project to be how can we better collect, organize, and disseminate HSV patient data for better understanding of HSV and patient experience? A secondary research question for the alpha phase is how can a patient registry be used as a complement to clinical trial data to evaluate the effectiveness of treatments and vaccines for HSV? Our hypothesis is that through a standardised patient-powered patient registry, researchers can gather better quality data for systematically studying HSV epidemiology and evaluate the effectiveness of treatments and vaccines.

## 4. Discussion

### 4.1. Principal Findings

Principally, this paper highlights the need for and advantages of an HSV patient registry and reports an in depth understanding of potential users of an HSV registry and their contexts, the technological constraints and the potential improvements that can be made. The design requirements and user stories for the HSV registry have been identified for further alpha phase development.

### 4.2. Strengths and Weaknesses

The strengths of the proposed features and design of this HSV patient registry are the patient-centricity that addresses user requirements and its interoperability with other systems, providing opportunities for the efficient integration of data. The key challenge for this registry will be to balance privacy, anonymization, and data security with the provision of open-access data for research and drug development. Further consideration in the alpha phase is needed to provide workable plans for engaging all ages and socio-demographic subgroups with digital registries and to overcome stigma and privacy concerns that might make patients less willing to share their data with clinicians and researchers. Variability in adoption of between socio-demographic groups was identified as a limitation of a number of registries [22], which must be addressed to minimise the risk of selection bias. Ensuring that these constraints are overcome will aid in enhancing the quality and efficiency of data collection that could contribute to the enhanced pharmacokinetic understanding of HSV treatments.

Privacy is a key concern in the current climate with particular concern as to how and where health information might be shared. Uncertainty or lack of transparency around the use of data can lead patients to adopt privacy-protective behaviours such as lying and not seeking care which would be detrimental to data quality and collection. As such, anonymisation was identified as one of the critical instruments for providing a secure environment for data sharing. Complete anonymisation; however, may limit patient access to their own data and/or the utility of the registry. As such, pseudo-anonymisation techniques will be investigated.

The investigated frameworks for the development of the HSV registry could help to facilitate the sharing of data whilst maintaining security. Patient-centred registries explicitly collect data with a view to multiple uses and can be repurposed to support service improvement and scientific inquiry. This would be valuable when considering the integration of this data with genomic data to further pharmacogenomic understanding in the treatment of HSV. Open-source registries also facilitate this as their primary aim is to solve the problem of data integration and facilitate ease of sharing and extensive data mining of patient data. Open source registry systems address standardisation and interoperability by using distributed searches to comply with data protection requirements and preserve data sovereignty. The security-focused architecture illustrated by the Gaucher registry [29] would be valuable to the application of the open-source framework to an HSV registry due to the sensitive nature of the data that would be collected. The emphasis on security could reassure patients and help to obtain high-quality data. instances and measures for obtaining consent will be further investigated in the alpha phase

The sustainability of a registry is a key challenge faced by most patient registries [35]. Agility, high quality data, robust secondary objectives, retaining patient engagement and maintenance of funding are all key aspects to support the sustainability of patient registries [36]. The sustainability of the architecture of the registry has been outlined above to retain flexibility for adding new diagnostic measures and data sources; further consideration of sustainability with regard to patient engagement and relevance of data collection and analysis will take place in the alpha phase.

### 4.3. Implications of the Proposed HSV Registry

The potential implications of an HSV patient registry for the advance of pharmacogenomics and personalized medicine research and development are promising. It would provide a database for unsupervised machine learning and data science to detect patterns, relevant variables, and sub-groups of the population. It would also allow researchers to easily recruit specific sub-groups of the population for studies, and to analyze the relationships between different variables and clinical outcomes in HSV patients. Interoperability between the registry and EHRs would provide more standardized, higher quality data. Together, these would lead to a better understanding of the variability in HSV expression and outcomes in patients. Epidemiologic and pharmacogenomic data could be used in conjunction to develop new and targeted drugs and vaccines and/or pre-empt potential side effects for specific sub-groups of the population. The feasibility and acceptability of integrating data with EHRs will be explored in greater detail in the alpha and beta phases.

### 4.4. Future Research

As a next step, we will explore researcher, patient and clinician use cases, user journeys and data requirements in detail, and build hypotheses for the most effective data science/machine learning involvement, using a data-oriented R&D approach, and proceed to the alpha phase of prototyping and testing different solutions and designs to best meet user requirements. A cost–benefit analysis will be done on the Cerner platform as a potential means of integrating the HSV patient registry with EHRs.

## 5. Conclusions

This paper describes the need for an HSV patient registry and design implications of identified policy constraints and user challenge to ensure efficient collection of high-quality, robust patient-level data to contribute to advancements in pharmacogenomic understanding and personalised medicine in HSV. This would provide a database for the application of machine learning and data science to enrich the data on pharmacokinetic variables in the HSV patient population. This will be further developed in the alpha phase of the project.

## Figures and Tables

**Table 1 biomedicines-07-00100-t001:** Interview themes, challenges raised, and implications for the system and data collection design.

Theme	Challenges for the User	System and Data Collection Design Implications
Stigma and anonymity	Patient: Requires anonymity, privacy and discretion to share data due to the stigma surrounding HSV	Patient motivation and needs must be considered
		Data must remain private and ideally anonymous
	Researcher: The quality of data is negatively affected by the ability and willingness to provide data and participate in studies	Details must be provided as to the use of the data to maximise data quality
		Education must be provided to raise awareness of HSV
Selection bias problems	Researcher: Patients with HSV are diverse in their socio-demographic backgrounds but also in the manifestation of HSV, not limited to those who have frequent recurrences, complications, pain, or psychological ramifications	Selection bias must be overcome
		The registry must be easily accessible by a wide body of populations
		Age-related accessibility must be considered throughout development, e.g., the choice of a technology platform for data collection
Understanding treatment and outcome gaps	Patient: many unaware of support or treatments after diagnosis and are not registered in the healthcare system	Data must be obtained on the unseen, to identify gaps and enable machine learning and unsupervised pattern recognition
	Researcher: Relevant and reliable data must be accessible in a suitable format that will help to inform and support research. There are gaps in current HSV treatment, management, or outcomes	A data solution should take into consideration the current gaps which might be affected by improved data collection
Risk factors and transmission	Patient: unsure how to alter their lifestyle to help minimise or mitigate recurrences	Consider associated lifestyle factors and ways to collect this data
	Researcher: Lifestyle factors play an important role in the spread and management of HSV	Enable enrichment and integration with multiple data sources (e.g., mobile applications)
Individualised vs. population-level	Researchers: The data needs to be integrated and enriched	Interoperability and importance of standardised data dissemination must be considered
		Link and enrich with EHR data
		Adhere to widely accepted data formats

**Table 2 biomedicines-07-00100-t002:** Researcher user personas and use cases.

**Researcher User Personas**	
Background	HSV research includes all medical research that attempts to prevent, treat or cure herpes, as well as fundamental research about the nature of herpes
Demographics	Education: Masters or above
Identifiers	Meticulous, require information in the database to be standardised with established governance and oversight plans
Goals	Good quality data that is standardised for meta-analysis
	Recruit patients for clinical trials
	Develop therapeutics or learn about population behaviour patterns and their association with disease development
	Improve or monitor health care
Challenges	Do not have access to the population of HSV patients for data collection
Objectives of the registry	Collect data from HSV patients in a standardised and accurate manner which can be centrally available
**Use Cases**	
Basic flow	Researchers conduct a literature search on HSV causes and relationships with patient behaviour
	Researchers hypothesize a potential relationship between certain activity and HSV for a certain patient group
	Researchers search the database to search for a certain rate of recurrence for a specific group of patients
	Termination outcome: Researchers use statistical methods to analyse database records to identify potential correlation between patient behaviour and disease
Alternative flow	Researchers want to identify certain patient groups for clinical trial recruitment
	Researchers look through a searchable database for patients that fit the clinical trial criteria
	Researchers contact the patients who gave consent.
	Termination outcome: Researchers identify suitable patients quickly

**Table 3 biomedicines-07-00100-t003:** Feature requirements derived from the use cases developed.

Use Cases/Requirements	Functionality	Description
Researcher and patient data access	Reports	Reports, based on the data in the system for the centre, can be generated in real-time. Graphs and tables can be visualised online
Patient permissions	Consent form	Ensure the patient has choice and control over their data
Researcher finds patient matching certain inclusion parameters	Search	Access to aggregated, anonymised, or pseudo-anonymised data
Longitudinal data	Follow up mechanism	A non-intrusive mechanism for follow-ups
		Give choice for being contacted, select frequency and reason
Patient registration	Registration	Pseudo-anonymisation
Researcher accessing user data	Clinical trial recruitment	Pseudo-anonymised user data list and communication
Addressing selection bias	Dynamic landing pages based on patient source	Different calls to action in recruitment channels (especially online) can be reinforced by custom landing pages
Longitudinal data	AppleHealth integration (on mobile devices)	Connector and UI

**Table 4 biomedicines-07-00100-t004:** Requirements for HSV registry design.

Category	Requirement/Issue	HSV Medical Registry Design Implications
Interoperability, population-level data	Data exchange	• API [29], FHIR [26]
		• Consider open-source platforms
		• Consider CERNER FHIR integrations
	Data analysis	Data collection, processing and analysis
	Common terminologies	Develop the system using the common and accepted standards and terminology for data schema definitions:
		International Classification of Diseases and its clinical modifications
		International Classification of Primary Care
		Medical Dictionary for Regulatory Activities (MedDRA)
	Cross-border integrations	PARENT [20]
Accessibility	Selection bias	Accessibility according to standards (across socio-demographic, geographical location, language groups, also split by familiarity with technology, ability to communicate, etc.)
Patient-centricity, privacy, patient goals and engagement	Roles of stakeholders are not defined	Define the roles of stakeholders via use cases [31]
Security	Patient access to data and content	Design user-friendly dashboards, updated with real-time information [16]
	Legislative requirements	Ensure GDPR, HIPAA compliance and on the EU level, adhere to the cross-border healthcare directive (CBHD). Consider anonymising the data before it is shared
	Personal privacy	Investigate pseudo-anonymization and interviews cases
	User experience (trust and openness)	Use common frameworks and templates [32] for:
		Design principles
		Look and feel—grid, colours and typography
		Reusable components and design patterns that solve common problems
		Content style guide—how to write
		Accessibility
	Security of technology	Consider encryption, server location, SSL, Database
Sustainability and extendability	Maintenance of the technology platform and operations	Holistic strategy for the system implementation, support and development
		No dependency on proprietary tech platforms (open-source, widely adopted tech). Low dependence on future tech maintenance [33]
	Long-term sustainability and development	Flexibility in allowing additional fields if there are new diagnostic methods
		Architecture allowing adding new data sources
		Low-maintenance technology and architecture

**Table 5 biomedicines-07-00100-t005:** Problem definition using the PICO framework.

Problem	Heterogeneous Dataset for HSV→Difficult to Analyse and Insufficient Understanding of HSV and Associated Diseases
Intervention	Primary: HSV patient registrySecondary: interoperable with EHR
Comparison, control or comparator	Non-HSV-specific registries
Outcome	Primary: Making use of unsupervised machine learning and data science methods employing quality and searchable dataset to allow researchers to analyse HSV patient data and recruit patientsSecondary: interoperable with other data sources and EHR

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
