# Peer review of "Engineering Requirements of a Herpes Simplex Virus Patient Registry: Discovery Phase of a Real-World Evidence Platform to Advance Pharmacogenomics and Personalized Medicine"

_biomedicines, 2019, doi:10.3390/biomedicines7040100_

Round 1

Reviewer 1 Report

This paper describes all steps that need to be undertaken to design and set up a HSV patient registry in a very elaborate and very clear manner. The prposed framework would be very helpful for other future investigators wishing to set up a comparable registry. The articles referenced in this paper are adequate and also very useful for further research.

I have a few minor comments:

the proposed set up for the registry is such that it can connect to any existing EHR. However, it would be worthwhile to mention that this process of integration with EHR's could be much simplified when the EHR adheres to modern IT standards for data exchange.  line 174: "the system uses API-based..": which system is meant here? line 178:'The registry .. will follow the National Standards for Electronic Health Records .." : where can I find information on this Standard? Please supply a reference. Line 185: " .. will be defined with a clear vision of real examples ..": the meaning of the whole sentence, and especially this part of the sentence is not clear to me. What will be defined? The rest of the manuscript is very concise and very clear written, in contrast to this sentence. Please rephrase.

Author Response

Dear Reviewer 1,
Attached find a point by point response to each of your helpful peer review comments.

With kind regards,
Edward

Reviewer 2 Report

I am the CEO of LymeDisease.org and the Principal Investigator of the MyLymeData patient registry.  The MyLymeData registry is a stand alone patient registry that was created by patients, is run by patients, and addresses issues that are important to patients.  I have also served on the Executive Committee of PCORnet, the big data project of the US Patient Centered Outcomes Research Institute, which, among other things included a large number of "patient-powered" registries.  I provide this background to put my comments in context.

I appreciated the scope of this publication and believe that the topic is important. Few articles address the issues of forming a patient registry.  The authors are from the UK, which has a different health care structure than the US. (Here data systems may be much more siloed and interoperability may be more of a problem.)

The description of the promise of patient registries in terms of contribution to science with phenotype of both patient and the disease and how subgroup analysis may be helpful is useful to enhance treatment effectiveness is quite good. I also agree that HSV is a critical area to focus on for all the reasons set forth quite articulately in the article.

I was left with many questions about the project that I thought the article should discuss or clarify.  For example, I am used to the phrase patient-powered meaning in the context of a partnership or patient leadership role.  Here, it seems that the expression is used to mean "laboring oar" without a power dynamic of patient involvement. It was not clear to me in reading this how patients were involved in your analysis of registry designs considerations to address patient needs. 

Did you conduct a focus group or survey of the patient population to determine their needs?  Do you intend to involve patients in the governance of the registry, determination of outcomes important to patients, or how to engage them in an on-going effort?  Will you use gamification of study results, for example with patients shown a graph of their responses compared to the responses of the total group, when they enter data?  Will they be able to download reports to share with other physicians? Are you planning on sharing results with the patients?  Will patient be involved in determining what research their data is going to be used in?  Will patient data be shared with pharma, will it be sold?  Will insurers have access to this data?

Unless you have significant engagement with patients, I think this is a researcher centric model of a patient registry. Given this and your identification as a challenge--"do not have access to a population with HSV", this project will likely be difficult to pull off successfully.  There is no community for those with HSV that I am aware of.  If that is true, you will need to build one.  How will you do that without patient engagement? Will you work with physicians?

What is the sustainability plan for this registry? This is usually the number 1 challenge of patient registries. How will it be funded? If it is a longitudinal study (as suggested) how will you drive continued patient engagement over time?  Will there be a patient forum or online private chat group?

The article does a great job of recognizing the issues related to stigma and privacy.  I appreciated the discussion of consent issues as well.  Is this going to be a one-time consent process or will patients consent to each study and use of the data?  Will they know who is using their data? 

Regarding stigma, I would note that "lifestyle factors and causes" --which is listed as a challenge for researchers in Table 1 is likely to raise "patient shaming" concerns among patients.  

I note that although the article is intended to address two personas--the patient and the researcher, yet a glance at table 1 shows a lot about researchers and very little (one item?) includes patients challenges.

You are interested in avoiding selection bias.  My experience in dealing with a disease that may be minor and acute only or which may be persistent with significant quality of life issues, is that those who sign up are those who have persistent on-going issues.  They need help.  Those for whom the disease is a minor inconvenience have no need for engagement. This is something you may want to consider for an opt-in (do I have that correct?) registry. My sense is that selection bias is inevitable and you may, accordingly, wish to target the registry toward that selection bias (those with more severe presentations) and acknowledge that that is an inherent sample limitation. 

On page 9, line 230 "The weakness are in balancing privacy, anonymization and data security with the provision of open-access data for researchers and drug development" --this is not a "weakness", it is a challenge.  It is an enormous challenge.  We do this through community based data stewardship. As a trusted 30 year old patient community that engages regularly with patients and is comprised of patients, we determine which research projects proposed align with patient interests and will be for their benefit--given concerns about stigma etc.  I don't think a group of researchers can do this because they cannot represent patient interests. I encourage you to engage with patients in a more robust manner.

You expect to work with the Cerner EHR platform. In the US, none of the big EHR systems is interested in sharing data collected on their platforms.  The UK may have more clout with EHR vendors and interoperability?  It is a major stumbling block in the US and the ONC (Office of the National Coordinator for Health IT) has proposed data sharing regulations that put the patient at the center of health care data sharing to address the issue of siloed health care data controlled by EHR platforms.  You may want to assess the feasibility of the EHR cooperability on this matter.  In the US, they regard data collected as belonging to them to use as they see fit--some sell to pharma, for example.

I hope these comments are useful to the authors.

Author Response

Dear Reviewer 2,

Attached find a point by point response to each of your peer review comments.

With kind regards,

Edward
